# Patient-physician communication about financial problems: A cross-sectional study among over-indebted individuals

Jacqueline Warth[ID][1]*, Marie-Therese Puth[1,2], Ulrike Zier[1], Niklas Beckmann[1], Johannes Porz[1], Judith Tillmann[1], Klaus Weckbecker[1,3], Hans Bosma[4], Birgitta Weltermann[1], Eva Münster[1]

1 Institute of General Practice and Family Medicine, University of Bonn, Bonn, Germany, 2 Institute for Medical Biometry, Informatics and Epidemiology (IMBIE), University Hospital Bonn, Bonn, Germany, 3 Institute of General Practice, Medical Faculty of the Heinrich-Heine-University Düsseldorf, Düsseldorf, Germany, 4 Department of Social Medicine, Care and Public Health Research Institute (CAPHRI), Maastricht University, Faculty of Health, Medicine and Life Sciences, Maastricht, The Netherlands

* Jacqueline.warth@ukbonn.de

**Data Availability Statement:** Data on over-indebted individuals (OID survey) cannot be shared publicly, as it contains potentially identifying participant information that could compromise

## Abstract

### Background

About every tenth household across Europe is unable to meet payment obligations and living expenses on an ongoing basis and is thus considered over-indebted. Previous research suggests that over-indebtedness reflects a potential cause and consequence of psychosomatic health problems and limited access to care. However, it is unclear whether those affected discuss their financial problems with general practitioners. Therefore, this study examined patient-physician communication about financial problems in general practice among over-indebted individuals.

### Methods

We conducted a cross-sectional survey among clients of 70 debt advice agencies in North Rhine-Westphalia, Germany, in 2017. We assessed the prevalence of patient-physician communication about financial problems and its association with patient characteristics using descriptive statistics and logistic regression analysis. Of 699 individuals who returned the questionnaire (response rate:50.2%), we included 598 respondents enrolled in statutory health insurance with complete outcome data in the analyses.

### Results

Conversations about financial problems with general practitioners were reported by 22.6% (n = 135) of respondents. Individuals with a high educational level were less likely to report such conversations than those with medium educational level (aOR 0.11; 95%CI 0.01–0.83) after adjustment for other sociodemographic characteristics, health status and measures of financial distress. Those without a migrant background(aOR 2.09; 95%CI 1.32–3.32), the chronically ill(aOR 1.90; 95%CI 1.16–3.13) and individuals who reported high financial

participants' privacy. Data access requests may be sent to the ethics committee of the Medical Faculty of the University of Bonn (Email: ethik@unibonn. de).

**Funding:** - Project manager (EM), OID survey - grant number n.a. - Landeszentrum Gesundheit NRW - Centre for Health North Rhine-Westphalia, Germany - https://www.lzg.nrw.de/ - The funders had no role in study design, data collection and analysis, decision to publish, or preparation of the manuscript.

**Competing interests:** The authors have declared that no competing interests exist.

distress(aOR 2.15; 95%CI 1.22–3.78) and cutting on necessities to pay for medications (aOR 1.86; 95%CI 1.12–3.09) were more likely to discuss financial problems than their counterparts.

## Conclusions

Few over-indebted individuals discussed financial problems with their general practitioner. Patients' health status, coping strategies and perception of financial distress might contribute to variations in disclosure of financial problems. Thus, enhancing communication and screening by routine assessment of financial problems in clinical practice can help to identify vulnerable patients and promote access to health care and social services and well-being for all.

## Introduction

General practitioners (GPs) are often the point of first medical contact for health problems within health care systems in Europe and have a coordinating role in many countries [1]. In line with research on the social determinants of health [2, 3], social factors are part of day-to-day clinical practice. However, the prevalence of social problems such as financial difficulties among patients, let alone communication about these problems, in the general practice setting has yet been understudied.

Over-indebtedness is widespread in Europe [4]. Currently, 6.9 million individuals in Germany alone face over-indebtedness which implies being unable to meet payment obligations and cover living expenses with available income and assets on an ongoing basis [5]. Recent studies have drawn attention to over-indebtedness as a potential cause and consequence of poor health [6]. Studies found an association between over-indebtedness and poor health outcomes that was not explained by standard socioeconomic status (SES) measures such as income and education [7–12]. A 15-year longitudinal study among 48778 adults in Finland found an association between over-indebtedness and an increased incidence of various chronic diseases including diabetes and psychoses [13]. These findings suggest that over-indebtedness may reflect a distinct risk factor of poor health. Furthermore, cost of illness can adversely affect health outcomes and access to medical services, and may ultimately, result in increased use of health care [14, 15]. Studies suggest that particularly vulnerable patient groups such as those with a low income, lack of health insurance or debt have an increased risk of cost-related medication non-adherence (CRN) [16–21] or forgone care [22]. Most health systems across high-income countries such as Germany impose cost sharing for health services [23].

Thus, prior research indicates that over-indebtedness is likely to reflect not only financial but also considerable health-related problems. In line with a number of studies patient-physician communication might generally contribute to improved health outcomes [24, 25]. More specifically, patient-physician communication about financial problems may help to prevent limited access to health care and poor health outcomes among those at risk. General practitioners may assist patients with health-related financial problems by a variety of strategies including reducing out-of-pocket costs or referral to social services [26–29]. However, little is known about whether and how patients and physicians discuss financial problems in general practice.

Estimates of the frequency of financial problems among patients in general practice vary considerably by population characteristics and measures used [30, 31, 32]. In a survey among 489 general practitioners in Germany, the majority of GPs (53.4 percent) reported that they were consulted by patients with poverty and/or financial problems at least three times a week [33]. Prevalence of cost conversations assessed by surveys among patients ranged from 16 percent in a US sample of 4050 chronically ill adults aged 50 years or older [34] up to 61 percent of elderly Medicare beneficiaries who reported cost-related medication non-adherence [26]. Findings are mixed as to what role patient and physician sociodemographic characteristics and patients' health status play in patient-physician communication related to financial issues [33, 34–40]. Nevertheless, studies have consistently found that patients have a desire to have cost conversations with their physician, yet many patients never had these conversations [26, 41, 42].

It is important to advance the understanding of patient-physician communication about social problems, including over-indebtedness, to promote health and access to health care for all. Therefore, the aim of this study is to assess the frequency of patient-physician communication about financial problems among over-indebted individuals in Germany, and to identify patients' characteristics that are associated with discussion of such concerns in general practice.

## Methods

### Data

This cross-sectional survey among clients of debt advice agencies examined health, medication use and self-medication in the over-indebted (OID survey; German acronym: ArSemü) [43]. Between July and October 2017, 70 of 145 approved debt advice agencies throughout the German federal state of North Rhine-Westphalia (NRW) conducted the recruitment of participants. Debt advice agencies that provide debt and insolvency counselling services to over-indebted consumers in Germany (Insolvency Statute; German: Insolvenzordnung; §305) were invited to act as recruiters by their umbrella organisation, namely the local German Consumer Organisation or the 'Expert Committee Debt Counselling of Non-statutory Welfare NRW' (German: Fachausschuss Schuldnerberatung der Freien Wohlfahrtspflege NRW) [44]. Each debt advice agency that agreed to participate received a specific amount of study material that corresponded to the number of clients and advisors at each site identified a priori (mean 24; min. 5 to max. 100 questionnaires). Data was collected by a self-administered written questionnaire returned to the study centre by mail: Clients received an anonymous standardised questionnaire and a postage-paid return envelope from their counsellor after the consultation when these were identified eligible according to the following criteria: a) completed at least an initial consultation based on the premise that it reflects a sensitive situation necessary to build trust; b) minimum age of 16 years due to contractual capability; c) sufficient language, reading and writing skills owing to the data collection method; d) one participant per household.

### Variables

The outcome measure was patient-physician communication about financial problems in general practice. Participants self-reported whether they had ever discussed their financial situation with their regular general practitioner (yes; no) when they reported to have a GP, they first consult in case of health problems (yes; no). Sociodemographic and health factors as well as participants' financial distress due to debt were considered in the analysis.

Sociodemographic information included sex, age, educational level, employment status, migrant background, marital status and number of children to account for general differences

in patient-physician communication patterns [45–52]. We classified age into three age groups (18–29; 30–49; 50–79 years). Self-reported data on the highest general educational and vocational qualifications were classified into three levels of education using the International Standard Classification of Education (ISCED) [53]: We distinguished between low (primary education, lower secondary education), medium (upper secondary education, post-secondary non-tertiary education) and high educational levels (tertiary education). Participants that reported full, part-time or marginal employment were classified as employed. In our study, we assumed that those with a lower SES (low educational level; unemployment), on the one hand, might feel compelled to discuss financial problems with their GP or have limited communication abilities that prevent such dialogue, on the other hand. Those with a higher SES might be especially reluctant to disclose financial problems due to feelings of shame, and may adopt different coping strategies. A migrant background was assumed when participants or at least one parent were born outside of Germany. Factors such as language barriers or differences in beliefs about illness and patient-physician interaction may hinder communication about financial problems in those with a migrant background [47, 52]. We classified participants' marital status into three groups: married, previously married (divorced or widowed), and never married. Number of children was classified into three groups (no children; 1 child; 2 or more children). The latter two variables were taken into account to examine patients' social support linked to marital status and household living expenses that vary by the number of children, and may influence communicative behaviour and interaction with their GP.

Patients' needs and expectations that can influence patient-physician communication might also depend on health status, disease stage and course of treatment [49]. Therefore, we considered both chronic diseases and recent visit to a general practitioner in the statistical analysis. Participants self-reported any chronic health conditions (yes,—please specify; no; don't know) and medication use in the last seven days. Medical experts reviewed self-reported data on both chronic conditions and medication use (pharmaceutical, underlying health condition) to identify and categorize chronic diseases according to ICD-10-GM (German adaptation of the International Statistical Classification of Diseases and Related Health Problems). Participants reported the use of outpatient and inpatient health care in the previous 12 months, including a visit to the GP (yes; no).

Moreover, we assumed that a high degree of financial distress increases individuals' perceived need for communication about financial problems. Thus, we included the following measures in the analysis to account for the degree of stress related to debt and cost of illness as well as patients' strategies to cope with their financial problems: The level of self-reported subjective financial distress due to debt was measured on a five-point Likert-scale which was dichotomized to distinguish low (not at all; somewhat; moderately) and high financial distress (to a great extent; to a very great extent). In Germany, adults enrolled in statutory health insurance need to pay co-payments for health services (German Social Code Book V § 61). For instance, co-payments for in-patient care amount to ten euros per calendar day, ten percent of costs for each prescribed medication (min. five, max. ten euros), and an additional fee of ten euros per prescription of therapies such as physiotherapy, speech therapy or occupational therapy [54]. Therefore, data on self-reported strategies to cope with health-related expenses were collected, including cost-related medication non-adherence and cutting on necessities to pay for medications (yes; no) in the previous 12 months was assessed. More specifically, the questionnaire captured CRN behaviours such as delaying or not filling prescriptions, skipping or decreasing doses of prescribed medications for financial reasons (yes; no).

## Statistical analysis

Descriptive analyses were performed to examine the prevalence of patient-physician communication about financial problems and characteristics of over-indebted individuals who discussed financial problems with their GP. Subsequently, multiple logistic regression analysis was used to assess the association between both sociodemographic and health factors, as well as measures of financial distress due to debt and patient-physician communication about financial problems (no; yes). All missing values within covariates were assigned to the most frequent response category when these were below the threshold of 5%. A separate response category was generated for missings in data on migrant background as these were above the predefined threshold. Covariates were entered into the model simultaneously. Statistical significance level was set at alpha = 0.05. We performed sensitivity analysis using complete case data to validate this approach. Analyses were carried out using IBM SPSS Statistics (version 25).

## Results

Of 1393 clients that were invited to participate in this study by debt advisors, 699 subjects returned the questionnaire with complete data on sex and age (response rate: 50.2%). We excluded participants who had a private health insurance (n = 7) or no health insurance (n = 2), and those who provided no information on insurance (n = 25), did not report to have a GP and/or provided no outcome information (n = 74). Characteristics of all participants included in the analysis (n = 598), stratified by patient-physician communication about financial problems are shown in Table 1.

Male (43.5%) and female (56.5%) participants were included in the analyses in nearly equal shares. The mean age of all subjects was 43.8 years (median 44.0; standard deviation ±13.0; minimum-maximum 19–76 years). Chronic diseases were widespread in the over-indebted sample (62.2%).

The prevalence of patient-physician communication about financial problems in general practice was 22.6 percent. Nearly a quarter of participants with a low (22.6%) and medium (24.4%) educational level have talked about this issue with their GP whereas only 3.6 percent of participants with a high educational level reported such communication. Among the chronically ill, 27.4 percent have discussed financial problems with their GP (14.0% in those without a chronic disease). In participants who reported high subjective financial distress, 25.9 percent discussed financial problems with their general practitioner. Among those who reported CRN, hence did not fill a prescription or skipped or decreased doses of prescribed medication due to financial problems in the last 12 months, less than a third of participants discussed this issue in general practice (28.2%) whereas such communication was more frequent in participants who reported to have recently cut on necessities to pay for medications (35.4%).

Multiple logistic regression analysis (Table 2) showed that patient-physician communication about financial problems was associated with over-indebted individuals' sociodemographic characteristics, health factors and measures of financial distress. After adjusting for other covariates, those with high educational level had significantly lower odds of self-reported communication about financial problems with their general practitioner than those with medium educational level (aOR 0.11; 95% CI 0.01–0.83). Individuals without a migrant background had greater odds of communication about financial problems than those with a migrant background (aOR 2.09; 95% CI 1.32–3.32). Other sociodemographic characteristics including sex, age, employment status, marital status and number of children were not associated with patient-physician communication about financial problems. The chronically ill had significantly higher odds of reporting such a conversation than those without a chronic disease (aOR 1.90; 95% CI 1.16–3.13). Bivariate analysis found a significant association between

**Table 1. Characteristics of participants (OID survey, n = 598).**

| | Communication about financial problems[†] | | | | | |
| | Total (n = 598) | | Yes (n = 135) | | No (n = 463) | |
| Variables | n | Col % | n | Row % | n | Row % |
|---|---|---|---|---|---|---|
| *Sociodemographic variables* | | | | | | |
| Sex | | | | | | |
| Male | 260 | 43.5 | 54 | 20.8 | 206 | 79.2 |
| Female | 338 | 56.5 | 81 | 24.0 | 257 | 76.0 |
| Age | | | | | | |
| 18–29 years | 99 | 16.6 | 16 | 16.2 | 83 | 83.8 |
| 30–49 years | 297 | 49.7 | 71 | 23.9 | 226 | 76.1 |
| 50–79 years | 202 | 33.8 | 48 | 23.8 | 154 | 76.2 |
| Educational level | | | | | | |
| Low | 266 | 44.5 | 60 | 22.6 | 206 | 77.4 |
| Medium | 303 | 50.7 | 74 | 24.4 | 229 | 75.6 |
| High | 28 | 4.7 | 1 | 3.6 | 27 | 96.4 |
| Missing | 1 | 0.2 | 0 | 0.0 | 1 | 100.0 |
| Employment status | | | | | | |
| Employed | 303 | 50.7 | 65 | 21.5 | 238 | 78.5 |
| Unemployed | 283 | 47.3 | 65 | 23.0 | 218 | 77.0 |
| Missing | 12 | 2.0 | 5 | 41.7 | 7 | 58.3 |
| Migrant background | | | | | | |
| Yes | 212 | 35.5 | 33 | 15.6 | 179 | 84.4 |
| No | 352 | 58.9 | 92 | 26.1 | 260 | 73.9 |
| Missing | 34 | 5.7 | 10 | 29.4 | 24 | 70.6 |
| Marital status | | | | | | |
| Married | 134 | 22.4 | 26 | 19.4 | 108 | 80.6 |
| Previously married | 234 | 39.1 | 58 | 24.8 | 176 | 75.2 |
| Never married | 223 | 37.3 | 50 | 22.4 | 173 | 77.6 |
| Missing | 7 | 1.2 | 1 | 14.3 | 6 | 85.7 |
| Number of children | | | | | | |
| No children | 171 | 28.6 | 40 | 23.4 | 131 | 76.6 |
| 1 child | 138 | 23.1 | 33 | 23.9 | 105 | 76.1 |
| 2 or more children | 285 | 47.7 | 61 | 21.4 | 224 | 78.6 |
| Missing | 4 | 0.7 | 1 | 25.0 | 3 | 75.0 |
| *Health status* | | | | | | |
| Chronic disease | | | | | | |
| No | 200 | 33.4 | 28 | 14.0 | 172 | 86.0 |
| Yes | 372 | 62.2 | 102 | 27.4 | 270 | 72.6 |
| Missing | 26 | 4.3 | 5 | 19.2 | 21 | 80.8 |
| Visit to GP, 12 months | | | | | | |
| No | 112 | 18.7 | 17 | 15.2 | 95 | 84.8 |
| Yes | 485 | 81.1 | 118 | 24.3 | 367 | 75.7 |
| Missing | 1 | 0.2 | 0 | 0.0 | 1 | 100.0 |
| *Financial distress* | | | | | | |
| Subjective financial distress | | | | | | |
| Low | 144 | 24.1 | 18 | 12.5 | 126 | 87.5 |
| High | 437 | 73.1 | 113 | 25.9 | 324 | 74.1 |
| Missing | 17 | 2.8 | 4 | 23.5 | 13 | 76.5 |

*(Continued)*

**Table 1.** (Continued)

| | Communication about financial problems[†] | | | | | |
| | Total (n = 598) | | Yes (n = 135) | | No (n = 463) | |
| **Variables** | **n** | **Col %** | **n** | **Row %** | **n** | **Row %** |
|---|---|---|---|---|---|---|
| Cost-related medication non-adherence, 12 months | | | | | | |
| No | 410 | 68.6 | 82 | 20.0 | 328 | 80.0 |
| Yes | 188 | 31.4 | 53 | 28.2 | 135 | 71.8 |
| Missing | 0 | 0.0 | 0 | 0.0 | 0 | 0.0 |
| Cutting on necessities to pay for medications, 12 months | | | | | | |
| No | 472 | 78.9 | 90 | 19.1 | 382 | 80.9 |
| Yes | 113 | 18.9 | 40 | 35.4 | 73 | 64.6 |
| Missing | 13 | 2.2 | 5 | 38.5 | 8 | 61.5 |

[†]Patient-physician communication about financial problems in general practice.

visiting a general practitioner in the last 12 months (OR 1.79; 95% CI 1.03–3.13) and communication about financial problems which, however, did not remain significant after adjusting for other covariates. Individuals who faced greater financial distress due to debt were more likely to report that they have discussed financial problems with their general practitioner: High subjective financial distress was associated with 2.15-fold (95% CI 1.22–3.78) higher odds of reporting communication about financial problems compared to low financial distress among the over-indebted. Cost-related medication non-adherence (OR 1.57; 95% CI 1.05–2.34) was associated with patient-physician communication about financial problems in bivariate analysis but not after adjusting for other factors. Individuals who reported cutting on necessities to pay for medications in the last 12 months (aOR 1.86; 95% CI 1.12–3.09) were significantly more likely to report such a conversation with their general practitioner. Sensitivity analysis showed similar patterns of findings.

## Discussion

The findings of this study demonstrate that less than one in four over-indebted individuals ever discussed financial problems in general practice. Even among those who reported high subjective financial distress or cost-related medication non-adherence, less than one in three talked with their GP about financial issues.

In line with previous research on patient-physician communication about financial problems [26, 30, 32–34, 39, 41, 55], these results reflect a considerable communication gap among over-indebted individuals. Opportunities to discuss financial problems in general practice might have been missed by both patients and general practitioners. Previous studies suggested that reasons why patients do not talk about financial problems in general practice may relate to individuals' preferences and abilities to communicate as well as their expectations of the patient-physician relationship and prior experiences [37, 41, 42, 56, 57]. Some patients may not disclose financial problems to their GP if they seek advice from other medical or social services such as debt advice agencies instead. General practitioners may also fail to initiate such conversations due to time constraints and competing demands, discomfort or perceived lack of knowledge about solutions to patients' financial problems [56, 58, 59].

Although all participants in the present study were considered over-indebted, there were significant variations in patient-physician communication about financial problems by specific patient characteristics.

**Table 2. Logistic regression analysis for patient-physician communication about financial problems (OID survey, n = 598)†.**

| | OR | 95% CI | aOR | 95% CI |
|---|---|---|---|---|
| *Sociodemographic variables* | | | | |
| Sex | | | | |
| Male | Reference | – | Reference | – |
| Female | 1.20 | 0.81–1.78 | 1.08 | 0.70–1.66 |
| Age | | | | |
| 18–29 years | Reference | – | Reference | – |
| 30–49 years | 1.63 | 0.90–2.96 | 1.61 | 0.83–3.12 |
| 50–79 years | 1.62 | 0.86–3.02 | 1.38 | 0.67–2.88 |
| Educational level | | | | |
| Low | 0.91 | 0.61–1.34 | 1.00 | 0.65–1.53 |
| Medium | Reference | – | Reference | – |
| High | *0.12* | *0.02–0.86* | *0.11* | *0.01–0.83* |
| Employment status | | | | |
| Employed | Reference | – | Reference | – |
| Unemployed | 1.04 | 0.71–1.53 | 0.90 | 0.59–1.37 |
| Migrant background | | | | |
| Yes | Reference | – | Reference | – |
| No | *1.92* | *1.24–2.98* | *2.09* | *1.32–3.32* |
| Missing | 2.26 | 0.99–5.16 | *2.60* | *1.08–6.25* |
| Marital status | | | | |
| Married | Reference | – | Reference | – |
| Previously married | 1.35 | 0.80–2.26 | 1.13 | 0.65–1.96 |
| Never married | 1.20 | 0.71–2.04 | 1.23 | 0.66–2.29 |
| Number of children | | | | |
| No children | Reference | – | Reference | – |
| 1 child | 1.03 | 0.61–1.74 | 1.02 | 0.56–1.86 |
| 2 or more children | 0.89 | 0.57–1.41 | 0.89 | 0.50–1.59 |
| *Health status* | | | | |
| Chronic disease | | | | |
| No | Reference | – | Reference | – |
| Yes | *2.26* | *1.43–3.57* | *1.90* | *1.16–3.13* |
| Visit to GP, 12 months | | | | |
| No | Reference | – | Reference | – |
| Yes | *1.79* | *1.03–3.13* | 1.53 | 0.85–2.77 |
| *Financial distress* | | | | |
| Subjective financial distress | | | | |
| Low | Reference | – | Reference | – |
| High | *2.43* | *1.42–4.16* | *2.15* | *1.22–3.78* |
| Cost-related medication non-adherence, 12 months | | | | |
| No | Reference | – | Reference | – |
| Yes | *1.57* | *1.05–2.34* | 1.14 | 0.73–1.77 |
| Cutting on necessities to pay for medications, 12 months | | | | |
| No | Reference | – | Reference | – |
| Yes | *2.25* | *1.44–3.51* | *1.86* | *1.12–3.09* |

†Odds ratios (OR), adjusted odds ratios (aOR) and 95% confidence intervals (CI); italics indicate significant results at alpha = 0.05.

Over-indebtedness may affect any individual across the socioeconomic spectrum [60]. However, experiences of loss of status, stigmatization and feelings of shame that can arise from ongoing over-indebtedness [61] possibly hamper patient-physician communication. Such experiences might be particularly distressing for individuals with a high educational level, and in turn reflect a barrier to communication about financial problems with their general practitioner. Likewise, cultural variations in the perception of debt-related worries, shame as well as expectations of the patient-physician relationship might contribute to the significant differences in patient-physician communication about financial problems by ethnic origin [47, 52].

Moreover, this study found an association between chronic disease as well as subjective financial distress and cutting on necessities to pay for medications in the last 12 months, and patient-physician communication about financial problems after adjustment. Patients who are chronically ill may be more likely to discuss their financial problems linked to continuity of care as well as co-payments for necessary medical services. In Germany, about 90 percent of the population is enrolled in statutory health insurance which enables adults to apply for reimbursement or waiver of co-payments that exceed two percent of the annual household income. When a physician attests a chronic condition, this ceiling can be reduced to one percent (§ 62 German Social Code Book V). High self-reported subjective financial stress and cutting on necessities to pay for medications might reflect the severity of financial problems on the one hand, and individuals' willingness to disclose and proactively deal with their financial problems on the other hand.

An encouraging finding is that some over-indebted patient groups that possibly bear a particularly heavy burden regarding their health status and financial distress are more likely to communicate about their financial problems with their general practitioner. The predictors of patient-physician communication in general practice identified in the present study were in line with several prior patient surveys [34, 39, 40]. However, few studies have examined the link between social determinants and patient-physician communication about financial problems in clinical encounters yet, most of which assessed communication in diverse US patient groups [32–42]. This is the first explorative study of such communication in a population of over-indebted individuals in Germany. The findings highlight that communication about such concerns in clinical encounters is rare even among over-indebted patients who face an increased risk of disease and are particularly susceptible to cost problems with regards to co-payments required for health services in Germany. These results warrant further research to draw conclusions about underlying reasons for differences in communication about financial problems within the over-indebted population. More specifically, qualitative data may help to gain a better understanding of barriers to communication both for patients and physicians.

## Limitations

This study has several limitations. First, data on patient-physician communication about financial problems was self-reported. Thus, data might be subject to recall or social desirability bias and differ from actual behaviour. Second, those who visit a debt advice agency to seek help might be less likely to seek additional advice from their GP. Likewise, individuals who lack sufficient language skills to complete the questionnaire were not included in the present study but are likely to face barriers to communication with their GP. Therefore, the prevalence of patient-physician communication in the over-indebted population could be underestimated. Individuals who disclosed communicating about their financial situation with their general practitioner might have been more likely to participate in the study to communicate about their debt-related health problems. Moreover, only patients who had reported having a regular general practitioner were subsequently asked about their communication with their

GP. Due to the latter aspects, the prevalence of conversations about financial distress by the over-indebted in the primary care setting might be overestimated. However, this selection bias can be assumed to have a minor effect on results of multiple regression analysis.

Third, on the basis of the available data, it is not possible to identify reasons why patients or general practitioners chose (not) to discuss financial problems but previous studies have addressed this issue [34, 37, 58]. Prior studies identified various strategies used during consultation to deal with health-related expenses in general practice [28, 62]. It remains to be established to what extent the conversations examined in the present study reflect an effective pathway for over-indebted patients and their general practitioners to enhance health outcomes and course of treatment, or to address the overall causes and consequences of financial problems.

Several methodological limitations and country-specific legal consequences of over-indebtedness limit the generalizability of our findings. Nevertheless, the present study reveals a need to raise awareness of patients' financial problems among relevant stakeholders throughout Germany. These findings may also apply to similar health and legal systems because any over-indebted individual can be considered at increased risk of illness and may experience limited access to health care.

## Conclusion

Although several studies have demonstrated that over-indebtedness may reflect a cause and consequence of poor physical and mental health, few over-indebted individuals were found to communicate about their financial problems in general practice. When financial problems remain unvoiced in general practice, patients may underuse medication and suffer from preventable adverse health effects associated with financial strain. It is therefore crucial to increase awareness about pathways to seek advice among patients when facing financial problems. General practice may serve as an important focal point for vulnerable patient groups due to GPs' key role in initiating and managing treatment, preventive healthcare and rehabilitation, and in coordinating various health and social services. Therefore, further training for GPs to identify and communicate about patients' financial problems as well as to transfer knowledge about available strategies and local social services is required. More specifically, routinely assessing financial problems like over-indebtedness in general practice can help to identify patients at risk. Recognition of such non-medical problems may facilitate general practitioners' efforts to provide patients with affordable and effective health care according to their need, and to prevent psychosomatic health complaints and concerns related to cost of illness.

## Acknowledgments

Special acknowledgement is due to the staff at each debt advice agency in the Federal State of North Rhine-Westphalia, Germany, for their support in data collection and all study participants.

## Author Contributions

**Conceptualization:** Klaus Weckbecker, Eva Münster.

**Data curation:** Jacqueline Warth, Marie-Therese Puth, Niklas Beckmann, Johannes Porz, Judith Tillmann, Eva Münster.

**Formal analysis:** Jacqueline Warth, Hans Bosma, Birgitta Weltermann.

**Funding acquisition:** Ulrike Zier, Klaus Weckbecker, Eva Münster.

**Methodology:** Jacqueline Warth, Marie-Therese Puth, Klaus Weckbecker, Eva Münster.

**Project administration:** Jacqueline Warth, Ulrike Zier, Judith Tillmann, Eva Münster.

**Supervision:** Eva Münster.

**Validation:** Jacqueline Warth, Marie-Therese Puth, Niklas Beckmann, Eva Münster.

**Writing – original draft:** Jacqueline Warth.

**Writing – review & editing:** Jacqueline Warth, Marie-Therese Puth, Ulrike Zier, Niklas Beckmann, Johannes Porz, Judith Tillmann, Klaus Weckbecker, Hans Bosma, Birgitta Weltermann, Eva Münster.

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
