## [Decision Letter · Decision Letter 0]

11 Mar 2020

PONE-D-19-33728

PATIENT-PHYSICIAN COMMUNICATION ABOUT FINANCIAL PROBLEMS: A CROSS-SECTIONAL STUDY AMONG OVER-INDEBTED INDIVIDUALS

PLOS ONE

Dear Dr. Warth,

Thank you for submitting your manuscript to PLOS ONE. After careful consideration, we feel that it has merit but does not fully meet PLOS ONE’s publication criteria as it currently stands. Therefore, we invite you to submit a revised version of the manuscript that addresses the points raised during the review process.

We would appreciate receiving your revised manuscript by Apr 25 2020 11:59PM. To enhance the reproducibility of your results, we recommend that if applicable you deposit your laboratory protocols in protocols.io, where a protocol can be assigned its own identifier (DOI) such that it can be cited independently in the future. For instructions see: http://journals.plos.org/plosone/s/submission-guidelines#loc-laboratory-protocols

We look forward to receiving your revised manuscript.

Kind regards,

Ali Montazeri

Academic Editor

PLOS ONE

Journal Requirements:

2) We note that you have indicated that data from this study are available upon request. PLOS only allows data to be available upon request if there are legal or ethical restrictions on sharing data publicly. For information on unacceptable data access restrictions, please see http://journals.plos.org/plosone/s/data-availability#loc-unacceptable-data-access-restrictions.

3) Please ensure you have thoroughly discussed any potential limitations of this study within the Discussion section.

Reviewers' comments:

Reviewer's Responses to Questions

**Comments to the Author**

1. Is the manuscript technically sound, and do the data support the conclusions?

Reviewer #1: Yes

Reviewer #2: Partly

2. Has the statistical analysis been performed appropriately and rigorously? 

Reviewer #1: Yes

Reviewer #2: N/A

3. Have the authors made all data underlying the findings in their manuscript fully available?

Reviewer #1: Yes

Reviewer #2: No

4. Is the manuscript presented in an intelligible fashion and written in standard English?

Reviewer #1: Yes

Reviewer #2: No

5. Review Comments to the Author

Reviewer #1: Introduction should be summarized.

Please adhere to STROBE guidelines for improving the quality of reporting of your study. This could be found at the link of below.

https://strobe-statement.org/index.php?id=available-checklists

Discussion: it is suggested to discuss a link between communication problems and social determinants of health in discussion section and explain what your study adds to the body of literature in this important field.

Also please recommend directions for future researches on the topic including qualitative studies to deeply understand the causes of not discussing financial costs of medical services either by patients or by physicians.

Please check the manuscript for some grammatical and writing errors. For example, page3 line 72; nonadherence

Reviewer #2: PATIENT-PHYSICIAN COMMUNICATION ABOUT FINANCIAL PROBLEMS: A CROSS-SECTIONAL STUDY AMONG OVER-INDEBTED INDIVIDUALS

This study introduced an important issue focused on social economic status and aimed to investigate patient-physician communication about financial problems in general practice among over-indebted individuals. The study conducted among 598 respondents (50.2% response rate) enrolled in statutory health insurance. The prevalence of patient-physician communication about financial problems was 22.6. The author reported that individuals with a high educational level were less likely to report such conversations than those with medium educational level. I think missing to report some of main items make study difficult to understand that I introduced below:

General comments

-Non-response analysis is not provided. How did the selection bias assessed? This should be assessed and discussed in the manuscript.

-sample size calculation was missed.

-please elaborate how data gathering was done? as study reported that it was self-reported.

-why 70 advice agencies was selected? Is that all or selected how they were selected?

-how Clients were recruited to the study?

-title of table 1 should be re-write and revised, it was illegible.

-Some of variables like educational level, financial distress, chronic diseases, and etc. were unclear in measurements.

-Each table should be inserted in one page. Tables 1 and 2 were placed in one page and half of table 2 is placed in next page.

-how participant were estimated over indebted?

6. PLOS authors have the option to publish the peer review history of their article (what does this mean?). If published, this will include your full peer review and any attached files.

Reviewer #1: Yes: Marzieh Aaraban

Reviewer #2: No

---

## [Author Response · Author response to Decision Letter 0]

8 Apr 2020

Please find a detailed point-by-point reply in the separate file attached.

---

## [Editor Report · Decision Letter 1]

21 Apr 2020

Patient-physician communication about financial problems: a cross-sectional study among over-indebted individuals

PONE-D-19-33728R1

Dear Dr. Warth,

We are pleased to inform you that your manuscript has been judged scientifically suitable for publication and will be formally accepted for publication once it complies with all outstanding technical requirements.

With kind regards,

Ali Montazeri

Academic Editor

PLOS ONE
---

## [Editor Report · Acceptance letter]

23 Apr 2020

PONE-D-19-33728R1 

Patient-physician communication about financial problems: a cross-sectional study among over-indebted individuals 

Dear Dr. Warth:

I am pleased to inform you that your manuscript has been deemed suitable for publication in PLOS ONE. Congratulations! Your manuscript is now with our production department. 

With kind regards,

on behalf of

Professor Ali Montazeri 

Academic Editor

PLOS ONE